# DNA Fragmentation in Human Spermatozoa and Pregnancy Rates after Intrauterine Insemination. Should the DFI Threshold Be Lowered?

**DOI:** 10.3390/jcm10061310

**Published:** 2021-03-22

**Authors:** Anne Sofie Rex, Chunsen Wu, Jørn Aagaard, Jens Fedder

**Affiliations:** 1Centre of Andrology & Fertility Clinic, Odense University Hospital, 5000 Odense, Denmark; asr@aagaardklinik.dk (A.S.R.); cwu@health.sdu.dk (C.W.); 2Aagaard Gynecological Clinic, Skejby, 8200 Aarhus, Denmark; jaagaard03@gmail.com; 3Research Unit of Gynecology and Obstetrics, Faculty of Health, University of Southern Denmark, 5000 Odense, Denmark

**Keywords:** male infertility, spermatozoa, DNA fragmentation, intrauterine insemination, pregnancy

## Abstract

Sperm DNA fragmentation index (DFI) can be analyzed by a flow cytometric assay after treatment with acid and acridine orange. In this prospective, cohort study, the value of DFI was determined in a semen analysis collected before fertility treatment (baselineDFI) in 146 couples and during 1–3 intrauterine inseminations (IUI) in 211 couples (511 cycles). The pregnancy rate (PR)/cycle was 9.9% if baselineDFI was >10 and 21.7% if baselineDFI was ≤10, (*p* < 0.005). The live birth rate (LBR)/cycle was 5% if baselineDFI was >10 and 14.2% if baselineDFI was ≤10 (*p* < 0.005). PR/patient was 23.1% if baselineDFI was >10 and 45.5% if baselineDFI was ≤10 (*p* < 0.005). LBR/patient was 12.4% if baselineDFI was >10 and 34% if baselineDFI was ≤10 (*p* < 0.005). When isolating non-stimulated IUI cycles and couples with female age < 35, a significant difference in PR and LBR between couples with high DFI and low DFI was seen. Results suggest that DFI > 10 could advice against timed coitus and non-stimulated IUI cycles. Analysis for DFI performed before treatment provides information about PR and LBR after IUI.

## 1. Introduction

It is estimated that every sixth couple in the western world will experience a period of infertility and that a male factor is involved in 50% of these, either exclusively or in combination with a female factor [1]. One of the aspects of DNA fragmentation that makes it particularly interesting is that it seems to be independent of basic semen characteristics [2]. This means that it may raise novel information about male fertility potential. There are several assays used to describe DNA fragmentation in the spermatozoa. The flow cytometric assay is believed to be the most objective and robust way to analyse DNA fragmentation as it, among other advantages, provides an objective analysis of thousands of cells in a short period of time [3,4]. It was recently shown that it is possible to implement a modified protocol of one of the two most widely used flow cytometric assays for DNA fragmentation, the sperm chromatin structure assay (SCSA). This assay allows for a quick and robust measurement of a large quantity of samples to obtain a clinical relevant DNA fragmentation index (DFI) [5]. The same has been seen for the other flow cytometric assay for DNA fragmentation, called the terminal deoxynucleotidyl transferase nick end labelling (TUNEL) [6].

It is well known that DNA fragmentation is associated with the fertility potential in both humans and domestic and other animals [7]. A systematic review of the literature let Zini (2011) [8] to conclude that “sperm DNA damage is associated with lower natural, intra-uterine insemination (IUI), and in vitro fertilization (IVF) pregnancy rates”. The chance of pregnancy after IUI treatment has in several studies been found to be as low as 4–5% if the amount of DNA fragmentation was increased [9,10,11]. Only in one study was this association not found. Here an increased risk of spontanous abortion was seen [12].

One aspect that still needs further clarification is how the analysis can be used when planning a couple’s fertility treatment. This was evaluated in a recent review and meta-analysis by Sugihara et al. (2020) [13]. They confirmed that analysis for DNA fragmentation has most value for in vivo treatment. They found that couples where the male had a normal amount of DNA fragmentation were three times more likely to conceive after IUI treatment which underpins the importance of the analysis.

However, all papers published to date have examined the level of DNA fragmentation in the same semen sample as the one used for the insemination, e.g., [9,11]. The workflow in flow cytometric analysis encourages that samples are collected and frozen [3] which means that the insemination has been performed long before the result of the DNA analysis was known. In order for the analysis to pose any value in treatment of the fertility treatment there is a need to evaluate whether an analysis for DNA fragmentation performed prior to treatment can differentiate between couples with higher or lower chance of success from the chosen treatment. Several authors have previously raised this concern [13,14].

Furthermore, it was demonstrated to be difficult to establish a definite cut-off between normal and increased amounts of DNA fragmentation. In the early papers, cut-off was often set to be DFI = 27 or 30 [9,10,11]. Conversely, in some more recent papers lower cut points, e.g., DFI = 15 [12] or DFI = 20 or 10 [15] have been suggested.

The aims of the current study are to investigate (I) the relationship between DFI measured in the individual cycles and the outcome of IUI treatments, (II) the relationship between DFI measured before treatment and the outcome of the subsequent fertility treatment, and (III) if a novel DFI threshold could be added.

The novelty of the study is that it investigates the significance of analyzing DFI before the treatment is initiated and that it investigates the relationship between DFI and the chance of pregnancy in couples matching a real-life scenario in the fertility clinic.

## 2. Materials and Methods

### 2.1. Study Design

This is a prospective study among couples referred to fertility treatment by their own physician. Most often, they have tried to conceive for more than a year. After the initial consultation where female and male factors were investigated and evaluated, the fertility doctor decides first line of treatment. If it was decided that if the couple should be treated with intrauterine insemination, they were eligible for inclusion in the study. Data from the initial three IUI cycles have been collected.

The study is approved by the Regional Scientific Ethics Committee (S-20140212, Regionshuset, Damhaven 12, Vejle, Denmark). Informed consent forms were collected for all participants before collection. All data was collected in the RedCap database in order to meet the requirements for data protection. 

### 2.2. Study Participants and Location for Collection

Participants were recruited by couples referred to a private fertility clinic, Aagaard Fertility Clinic, Skejby, Denmark, for treatment of infertility. Information about age, tubal passage, smoking, alcohol intake, and BMI was included for the females. All female partners were between 18 and 45 years of age on day of inclusion. All males were over the age of 18. Data for smoking and alcohol intake were included for the males.

### 2.3. Sample Collection

The patients were instructed to keep two to five days of abstinence before analysis. After liquefaction, but within 1 h after ejaculation, samples were evaluated on basis of standard semen characteristics such as semen volume, sperm concentration, motility, and total number of spermatozoa. Immediately after, a sample of 100 µL of raw semen was collected in a 1.5 mL micro tube (Easy cap, Sarstedt, North Rhine-Westphalia, Nümbrecht, Germany), flash frozen and preserved at −80 °C until the analysis for DNA fragmentation was performed.

### 2.4. IUI Treatment

All couples were referred for treatment at the fertility clinic by their own physician. At the initial work-up, it was evaluated that the first line of treatment would be timed coitus or intra uterine stimulation. All patients had patency through at least one tube evaluated by hystero salpingo ultrasonography. The cycles were either spontaneous or stimulated with chlomiphen and/or follicle stimulating hormone (FSH). All inseminations were performed with a minimum of one follicle. The semen samples used for the insemination were collected after masturbation and analyzed within one hour after the ejaculation. The samples were purified by a 40/80% gradient (Sil-Select FertiPro NV, Belgium; Sydney IVF Gamete buffer, COOK Medical, Australia) gradient centrifugation and washed twice in Sydney IVF Gamete buffer (COOK Medical, Bloomington, IN, USA). The guideline at the clinic states that insemination preferably should be performed with five million spermatozoa. All females injected hCG to induce ovulation approximately 36 h before insemination/timed coitus. The purified spermatozoa were placed in the uterus using Emtrac A catheter (Gynétics, Belgium). A maximum of three cycles were performed per couple.

Pregnancy was reported by the patients and confirmed by the presence of gestational product by ultrasound in gestational week 7. After term, information of live birth was collected.

### 2.5. Principles for the DNA Fragmentation Assay

One of the most widely used techniques for an analysis of DNA fragmentation in spermatozoa is a flow cytometric analysis after a denaturation with acid and coloration with acridine orange (AO) [16]. This was initially denoted SCSA due to the specific flow cytometric software used. Later, it has been seen that other types of software, e.g., the FACSDiva 6.1.3 results in equally clinically relevant levels of DNA fragmentation [5]. The assay for sample preparation is identical with the SCSA.

The flow cytometric DNA fragmentation assay uses that the cell membrane permeable molecule AO interacts with both double stranded (ds) and single stranded (ss) DNA. When AO is intercalated with dsDNA it will emit green light when it encounters with the blue laser from the flow cytometer. AO intercalated with ssDNA will emit red light.

#### 2.5.1. Reagents and Utensils

The reagents needed for the analysis are an acid solution (20 mL 2.0 M HCl (0.08 M); 4.39 g NaCl (0.15 M); 0.5 mL Triton X-100 (1%); ddH_2_O to 500 mL; pH was adjusted to 1.2 med 5 M HCl) a 1× TNE buffer (9.48 g Tris-HCl (158 0.1 M); 52.6 g NaCl (F.W. 58.44 1.5 M); 2.23 g EDTA (disodium, F.W. = 372.24 10 mM); ddH_2_O to 600 mL; Ph was adjusted to 7.4 using 2 M NaOH. Before use, the 10× TNE buffer is diluted to 1× TNE buffer. Storage time up to 1 year at 5 °C) and a 0.015‰ AO staining solution comprised of AO (Polyscience, 400 Valley Road, Warrington, UK) and a coloring buffer (370 mL 0.1 M Citric acid; 630 mL 0.2 M Na_2_PO_4_ buffer; 372 mg EDTA (disodium, F.W. = 372.24 1 mM); 8.77 g NaCl (0.15 M). Mixed overnight for the EDTA to dissolve. pH is adjusted to 6.0).

#### 2.5.2. Reference Sample

For approximately every tenth sample a reference sample is analyzed. This is a sample with a known DNA fragmentation pattern that allows us to detect any irregularities or fluctuation in the analysis. It could also be used to compare flow cytometric DNA fragmentation analysis between laboratories.

#### 2.5.3. Sample Preparation

On the day of the analysis, the samples were thawed on ice and diluted in 5 mL tubes (Falcon, Reynosa, Mexico) with TNE buffer to a concentration of approximately six mio/mL in a total volume of 200 µL. All samples are diluted at least 1:1. Samples with a concentration below 12 mio/mL will also be diluted 1:1 and will thus have a lower final concentration. However, data must still be collected for 10,000 cells. The diluted samples were kept on ice (from the −20 °C freezer) and 400 µL acid solution were added to the samples. After exactly 30 s, 1.2 mL AO staining solution was added. Samples were left in the dark to equilibrate for 3 min.

#### 2.5.4. Flow Cytometric Analysis of DNA Fragmentation

After equilibration, cells were analyzed on a FacsCanto™ II flow cytometer (Becton, Dickinson and Company (BD), Franklin Lakes, NJ, USA) with a 488 nm blue laser (air-cooled, 20 mW solid state). When AO encounters the blue laser in the flow cytometer, AO intercalated with dsDNA will emit light at a maximum of 525 nm (green light) and when intercalated to ssDNA it will emit light with a maximum of 650 nm (red light). Based on 10,000 events from each analysis, the dot plots were created and analyzed by FACSDiva 6.1.3 (BD) which is a standard flow cytometric software. It is possible to use a prefabricated template; however, manual adjustment during analysis will be required. Further information about using the FACSDiva 6.1.3 software for determination of DNA fragmentation can be found in Rex et al. (2020) [5].

As the sample injection tube on the flow cytometer holds dead volume, it is recommended to run samples 30–60 s before recording. Samples should be analyzed on “low” and as far as possible not exceed a flow rate of 500 events per s. 

To obtain the best possible dot plot, minor adjustment of the voltage of the red and green parameter before recording might be advisable. Furthermore, adjustment of the gating when the analysis has been completed was also performed.

### 2.6. Statistical Analysis

The statistical analysis was performed using Stata 16 (StataCorp, College Station, TX, USA).

We fitted the logistic regression between pregnancy and DFI, and the DFI was modeled by cubic spline with 5 knots (4, 9, 12, 18.575, and 41.65), which are corresponding to the percentiles of DFI (5%, 27.5%, 50%, 72.5%, and 95% respectively). Afterwards, we plotted the odds of pregnancy according to DFI. Based on the plot, our best knowledge, and feasibility of interpretation, we dichotomized DFI as low (DFI ≤ 10) and high (DFI > 10). The cut point was used through the analyses.

We compared cycle-specific proportion of pregnancy among cycle-specific dichotomized DFI (DFI ≤ 10 and DFI > 10) among all cycles. We repeated the same analysis on each cycle (the first cycle, the second cycle, and the third cycle), with or without stimulation. Cycles with missing values were excluded.

Then we compared pregnancy proportion among DFI at baseline (DFI ≤ 10 and DFI > 10) per patient. We stratified the analysis according to with or without stimulation.

We restricted analysis to the patients in whom DFI was analyzed at least two times. Pearson correlation was calculated over the four measurements for DFI. The dichotomized DFI was further classified as three groups, consistently low (all DFI ≤ 10), consistently high (all DFI > 10) and mixed (some DFI ≤ 10 and some DFI > 10). We compared pregnancy among the three groups.

All the above analyses were performed by chi square test or by Fisher’s exact test, dependent on sample size. Stata 16 was used to implement the analyses. We combined our clinic knowledge and *p*-value < 0.05 to judge whether the results were clinically significant.

To adjust for potential confounders such as age, stimulation, total concentration of spermatozoa and concentration of motile spermatozoa, FSH and AMH levels of the female we used logistic regression to repeat the above analyses. BMI was excluded as a confounder as it seems not to impact fecundability after insemination [17].

It is well-known that DFI increases with the male age. The same goes for the female fertility potential. It could thus be speculated that a possible reduction in pregnancy rates in couples with increased DFI was merely an independent effect of correlating increased age within the couple. As the decline in female fertility potential accelerates after the age of 35, we included an analysis of couples where the female age was <35.

## 3. Results

A flow chart of participants and cycles included in the study can be found in Figure 1. Data for age (females), smoking and alcohol intake for participants were collected. Mean values for concentration of spermatozoa and motile spermatozoa collected at the initial semen analysis are presented in Table 1 and Table 2.

Very few patients smoke, and it is thus not meaningful to include this confounder. As regards to alcohol, only six females reported an intake of more than seven drinks/week. Despite some detrimental results, a vast majority of studies do not seem to find a correlation between a low intake of alcohol in females receiving fertility treatment and the chance of pregnancy [18]. Therefore, this confounder was excluded.

Pregnancy rates in each cycle were evaluated in relation to the male DFI analyzed from the semen sample used for the IUI treatment in that specific cycle. A total of 211 couples with 511 cycles were included. Additional 210 cycles were missing data. Of the 511 cycles, 209 were first cycles, 169 were second cycles, and 133 were third cycles. If DFI was ≤10 the chance of pregnancy was 13%, and if DFI was >10 the pregnancy rate was 9.4%, (*p* = 0.54) in the first cycle. When separating cycles in stimulated and non-stimulated IUI cycles a similar result was seen. In the second cycle, the difference in pregnancy rates between couples with low and high DFI was more pronounced, 15.6% for couples with DFI ≤ 10 and 8.9% for couples with DFI > 10 (*p* = 0.27). The difference was most pronounced in the non-stimulated cycles. In the third cycle, the difference between pregnancy rates for couples with DFI ≤ 10 and >10 was weak. When including data for live birth rate (LBR) similar results were seen. See Table 3.

### 3.1. Baseline DFI and the Chance of Pregnancy per Cycle from the Subsequent Fertility Treatment

We determined the pregnancy rates per cycle in patients with high or low baseline DFI, respectively. One to three stimulated or non-stimulated IUI cycles are included per patient. Data from 11 patients who have had the initial work-up and semen evaluation but spontaneous became pregnant before fertility treatment could be initiated were also included. A total of 146 couples were included with a total of 352 cycles. For couples where the male DFI was >10 the pregnancy rate was 9.9% compared to couples where the male DFI was ≤10 where the pregnancy rate was 21.7%, (*p* < 0.005). When adjusting for confounders the OR changed from 0.39 (95% CI 0.21–0.75) to 0.24 (95% CI 0.10–0.57). One-third of the cycles were non-stimulated cycles. In the non-stimulated cycles, the pregnancy rate was 25.5% if DFI was ≤10 at baseline and 6.1% if DFI was >10, (*p* < 0.005). The age of the female was on average 31 years of age. In couples where the female age < 35, the pregnancy rate was 9.9% for couples where DFI > 10 and 23.6 for couples where DFI ≤ 10 (*p* < 0.005). The results from LBR were similar. When adjusting for confounders the OR for LBR was further reduced from 0.32 (95% CI 0.15–0.69) to 0.20 (95% CI 0.07–0.56). See Table 4. For cycles where stimulation with either chlomiphen, FSH or both was administered the difference was non-significant.

### 3.2. Baseline DFI and the Chance of Pregnancy per Patient from the Subsequent Fertility Treatment

Analysis for DNA fragmentation in the semen sample from the initial evaluation was performed in 146 patients. Patients were included if they had 1–3 stimulated or non-stimulated cycles or a spontaneous pregnancy before treatment was initiated. Of these, 46 couples (31.5%) conceived. As above, pregnancy rates for couples with males with DFI > 10 or DFI ≤ 10 in the initial evaluation were calculated. The chance of pregnancy in 1–3 cycles was 45.5% if the male DFI was ≤10, and if the male DFI was >10, 23.1% of the couples conceived (*p* < 0.005). However, as the majority of the couples had both stimulated and non-stimulated cycles, it is not possible to adjust for this confounder. When adjusting for the remaining confounders the OR changed from 0.36 (95% CI 0.18–0.74) to 0.30 (95% CI 0.11–0.78). The LBR for couples where the initial evaluation of DNA fragmentation was ≤10 was 34% and if the level of DNA fragmentation was >10 the LBR was 12.4% (*p* < 0.005). When adjusting for the remaining confounders the OR changed from 0.27 (95% CI 0.12–0.64) to 0.14 (95% CI 0.04–0.49). The miscarriage rate for couples with low DFI is thus 39% and 48% if the couples with DFI > 10 (*p* = NS).

Data for couples with young female age (<35 years) showed that the pregnancy rate for these couples was 48.9% if the male DFI was ≤10 and 23.3% if the DFI was >10, (*p* < 0.005). The same was seen when including results from LBR. For all calculations, the OR was reduced when adjusting for confounders. See Table 5.

### 3.3. Fluctuation of DFI within the Same Individual

Data for DNA fragmentation was collected in 1–4 samples per male. For 197 of the included males, DFI was analyzed in more than one ejaculate. Data showed moderate correlation between the measurements with correlation coefficient between 0.42 and 0.64 depending on which ejaculates have been analyzed. For 62 males, it was found that DFI was low in one semen sample and high in another. The couples were divided into males with DFI ≤ 10 in all analyses, males with only DFI > 10 and patients with mixed values (DFI ≤ 10 and DFI > 10). For 126 of the males, the partner received fertility treatment. The pregnancy rate for couples, where the male had a consistently low DFI, was 36.8%. For couples with mixed values or consistently high DFI, the pregnancy rates were 32.7% and 24.2%, respectively, (*p* = 0.51). See Table 6.

## 4. Discussion

Within this dataset, we found statistically significant reduction in pregnancy rates and LBR when DFI exceeded 10. Data showed that the relationship between DFI and the chance of pregnancy was more distinct in the first and second cycle than in the third cycle. It also showed that DFI measured at the initial semen analysis is relevant when evaluating the chance of pregnancy in the subsequent IUI treatments. The results were similar when data were analyzed per cycle or per patient and when pregnancy rate and LBR were considered. These results were not associated with the female age as similar differences in pregnancy rates and LBR were seen in couples with only a female age <35. The results were significant in non-stimulated cycles. Pregnancy rates and LBR were similar in couples where semen samples have had fluctuating DFI to couples with samples with consistently low DFI. This could indicate that a slight fluctuation does not necessarily impact the chance of pregnancy and live birth and implies that a second analysis for DNA fragmentation could be relevant if the first analysis is increased.

### Strength and Limitations

The strength of the study is that the DNA fragmentation analysis was performed before the patients started their treatment. Results showed that DFI analyzed at their initial semen evaluation was relevant for the result of the subsequent IUI treatment. An advantage of the study is that the results were not only statistically significant. The difference in pregnancy rates could also be considered predictive when planning the treatment with the couple.

Power calculation shows that mean pregnancy rate of 45% vs. 23% in the two groups and significance level of 0.95 and a power of 80% can be obtained by 72 subjects in each group. When means are 37 vs. 15.5 in the two groups a significance level of 0.95 and 80% power can be obtained when including 65 patients in each group. Even though the ratio in the compared groups is not 1:1, we managed to include a sufficient number of patients (146 couples) when comparing the pregnancy rates per patient of 45% vs. 23%. For LBR per patient, we managed to include 142 couples which is also sufficient to compare pregnancy rates of 37% vs. 15.5%.

Some of the included couples never initiated their fertility treatment. However, this was most often due to divorce, relocation or other events causing the couples to cancel or postpone their fertility treatment. It is a limitation of the cycle-specific analysis that we were not able to collect DFI for all cycles. There is the probability that results could have been more distinct if a DFI sample for analysis was collected in all cycles. However, there is no indication that the missing values are more pronounced within couples with low or high DFI, respectively. We thus consider that the missing values are unlikely to cause any selection bias. Another limitation of the study is the consideration about whether to include couples who obtained pregnancy after their initial work-up, semen evaluation, and inclusion in the study but before they started their fertility treatment. If these couples were excluded from the study, the results from the most fertile couples would not have been included, thereby weakening the result. A decision was therefore made that these results should not be excluded. However, results are not included in the cycle-specific data.

Advice concerning abstinence for 2–5 days was given to all patients. However, specific information about abstinence time before each sample delivery was not collected. Even though the literature is not in agreement of the impact on abstinence time on DNA fragmentation, this is a possible confounder for which we cannot adjust. 

Due to the procedure at the collection site, information on sperm morphology is not available. This is another possible confounder for which we cannot adjust. 

The semen sample used for the insemination should include at least 5 million. However, due to the rather large fluctuation of semen parameters, some inseminations might have been performed with a lower number of spermatozoa as there is a general wish to continue the treatment when the female has been stimulated.

Data showed that the difference between pregnancy rates from patients with high or low DFI was lowest after the second IUI treatment. Several underlying factors can affect a couple’s fertility potential. It is possible that couples who do not conceive in the first two treatments have underlying factors that become more significant than DFI when reaching the third IUI.

When cycles were divided by stimulated and non-stimulated cycles, the pregnancy rate per cycle was higher in cycles with no stimulation. Besides small groups, another reason could be that patients were not randomized to either stimulated or non-stimulated cycles. The decision to stimulate might be caused by irregular period or increased age. It is thus a possibility that patients inseminated in non-stimulated cycles might be more fertile resulting in higher pregnancy rates.

Even though the relationship between fertility and DNA fragmentation in general has been accepted, the utility of the analysis in the everyday work in the fertility clinic and the value when planning the treatment is still debated. Several reviews and meta-analyses have evaluated the clinical value of testing for DNA fragmentation and some have raised questions about the utility of the test. However, it is a common problem when discussing DNA fragmentation and infertility that evaluation of the value of the analysis is made partially or solely on IVF and ICSI cycles [19]. It was described that the greatest value of DNA fragmentation analysis is seen when it comes to predicting time to pregnancy or positive outcome after IUI treatment [13]. A study by [2] showed that there was a significant difference in DFI in couples conceiving within 0–3 months and couples conceiving within 4–9 months and even greater in couples with no pregnancy within the first 12 months. Several studies have shown that the analysis of DNA fragmentation has predictive value for the chance of pregnancy in first pregnancy planners and for the chance of pregnancy after IUI treatment which is significantly reduced if the amount of DNA fragmentation is increased [9,10,11,20]. A significant increase in miscarriage has also been seen in couples where the males have increased DNA fragmentation [21].

Concerning the SCSA and the lack of consensus of an optimal threshold, it is correct that a specific cut-off has not been established. This is probably due to the fact that the effect on DNA fragmentation on fertility seems to be a matter of a sloping decrease and not definite cut-off. It has been seen several times that the success after IUI starts to decrease if DFI exceeds 20% and reaches a success rate of 0–5% when DFI is above 30% [5,9,11]. From the above results, it is possible that the fertility potential might start to decline at even lower levels of DNA fragmentation. The DFI should be used as an analysis which adds information and lift the semen evaluation to more than a matter of number and motility by including an evaluation of internal quality [1].

The above data implies that an analysis for DNA fragmentation does have value when planning the subsequent in vivo treatment, and that it should be analysed as early as possible. In this study, we found that pregnancy rates were affected already when DFI exceeded 10% and that the implications of DFI on pregnancy rates were more significant among young females and in the group of patients choosing non-stimulated IUI treatment. This could indicate that a novel cut-off value could be added to the existing data within the field of sperm DNA fragmentation. If the patient presents with a DFI above 10%, it could be considered not to recommend timed coitus and non-stimulated cycles. Further studies, including a group of patients receiving only non-stimulated cycles could be initiated to confirm this. If DFI exceeds 20%, it could prompt further diagnostics. If DFI exceeds 30%, further diagnostics and relevant therapeutical approaches should be initiated and ICSI could be considered [22].

## Figures and Tables

**Figure 1 jcm-10-01310-f001:**
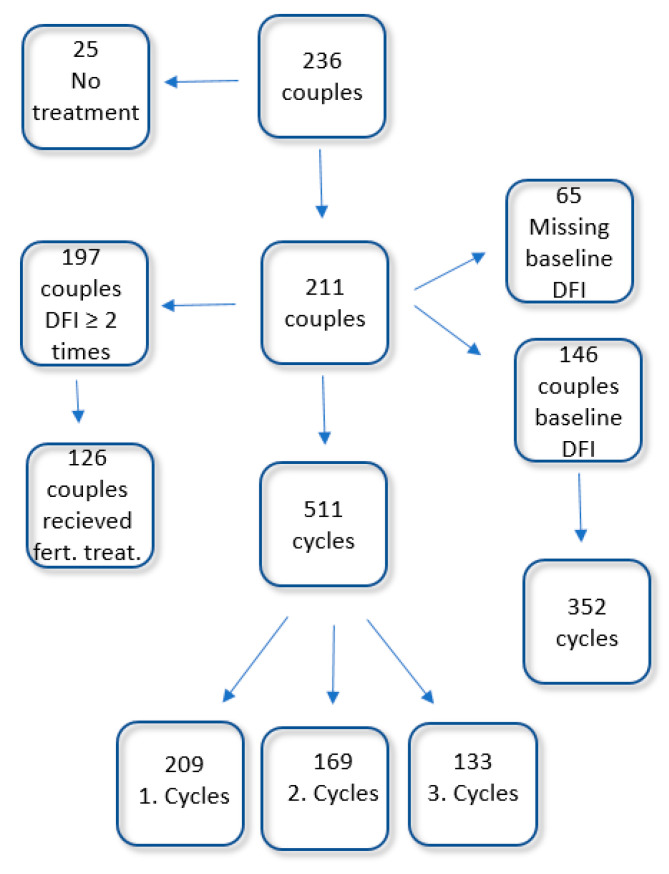
Of the 236 couples initially included in the study, 25 never received any form for treatment. For the remaining 211 couples, 511 cycles were included. For 146 of the patients, a baseline DFI was obtained. From these patients, 352 cycles were available. For 197 of the couples, DFI was analyzed in minimum two different samples. Of these, 126 couples received fertility treatment.

**Table 1 jcm-10-01310-t001:** Basic characteristics for all couples.

		Females Mean (SD)	Males Mean (SD)
Mean age		31.4 (4.6)	
BMI		24.2 (4.5)	
Missing		35	
Smoking	No	170 (80.6)	158 (78.9)
	Yes	13 (6.2)	22 (10.4)
	Missing	28 (13.2)	31 (14.7)
Alcohol	No	70 (33.2)	34 (16.1)
	Yes	112 (53.1)	146 (69.2)
	missing	29 (13.7)	31 (14.7)

**Table 2 jcm-10-01310-t002:** Basic characteristics for couples with baseline semen evaluation.

		Females Mean (SD)	Males Mean (SD)
Mean age		30.8 (4.6)	
BMI		24.4 (4.7)	
Missing		29	
Smoking	No	118 (80.8)	108 (74.0)
	Yes	4 (2.7)	13 (8.9)
	Missing	24 (16.4)	25 (17.1)
Alcohol	No	48 (32.9)	23 (15.8)
	Yes	72 (49.3)	98 (67.1)
	Missing	26 (17.8)	25 (17.1)
Motile sperm (mio/mL) (Il 25%/50%/75%)			32 (±27) (10/25/50)
Total spermatozoa (mio/mL) (Il 25%/50%/75%)			47 (±28) (20/35/75)

**Table 3 jcm-10-01310-t003:** Pregnancy rates and live birth rate for the first, second and third IUI is presented. All females received an ovulation inducer before insemination. Approximately half of the cycles were otherwise non-stimulated. The other half was stimulated with chlomiphen and/or FSH.

	1.IUI		2.IUI		3.IUI		Total		Crude OR (95% CI)	Adjusted OR (95% CI)
	(%)	*p*	(%)	*p*	(%)	*p*	(%)	*p*		
**Pregnancy Rate**										
Total										
DFI ≤ 10	13		15.6		14		14.2		0.79	0.67
DFI > 10	9.4	0.54	8.9	0.27	17	0.68	11.5	0.47	(0.41–1.47)	(0.28–1.61)
Non-stimulated cycles										
DFI ≤ 10	12.1		23.8		12.1		14.8			
DFI > 10	9.4	1	7.7	0.22	9.4	1	9.6	0.36		
Stimulated cycles										
DFI ≤ 10	13.9		11.6		13.9		13.9			
DFI > 10	9.5	1	10	1	9.5	1	13.1	0.89		
**Live Birth Rate**										
Total										
DFI ≤ 10	11.6		10.9		10		10.9		0.75	0.78
DFI > 10	5.9	0.35	7.1	0.54	12.8	0.67	8.4	0.44	(0.36–1.57)	(0.32–1.94)
Non-stimulated cycles										
DFI ≤ 10	9.1		14.3		9.1		9.8			
DFI > 10	6.5	1	73.9	0.31	6.5	1	6.9	0.55		
Stimulated cycles										
DFI ≤ 10	13.9		9.3		13.9		11.5			
DFI > 10	5	0.40	10	1	5	0.40	9.8	0.70		

**Table 4 jcm-10-01310-t004:** Baseline DFI was analyzed before treatment was initiated. The spontaneous pregnancies obtained between enrolment but before treatment was initiated was included in the total population but excluded in the groups with the stimulated vs. non-stimulated cycles.

	N Cycles	N Pregnancies	(%)	*p*-Value	Crude OR (95% CI)	Adjusted OR (95% CI)
**Pregnancy Rate**						
Total	352	50	14.2			
DFI ≤ 10 at baseline	129	28	21.7		0.39	0.24
DFI > 10 at baseline	223	22	9.9	<0.005	(0.21–0.75)	(0.10–0.57)
Stimulated cycles						
DFI ≤ 10 at baseline	79	13	16.5			
DFI > 10 at baseline	137	13	9.5	0.13		
Non-stimulated cycles						
DFI ≤ 10 at baseline	47	12	25.5			
DFI > 10 at baseline	82	5	6.1	<0.005		
Female age < 35						
DFI ≤ 10 at baseline	110	26	23.6			
DFI > 10 at baseline	182	18	9.9	<0.005		
**Live Birth Rate**						
Total	348	29	8.3			
DFI ≤ 10 at baseline	127	18	14.2		0.32	0.20
DFI > 10 at baseline	221	11	5	<0.005	(0.15–0.69)	(0.07–0.56)
Stimulated cycles						
DFI ≤ 10 at baseline	78	9	11.5			
DFI > 10 at baseline	136	8	5.9	0.14		
Non-stimulated cycles						
DFI ≤ 10 at baseline	47	9	19.2			
DFI > 10 at baseline	82	3	3.7	<0.01		
Female age < 35						
DFI ≤ 10 at baseline	109	17	15.6			
DFI > 10 at baseline	180	11	6.1	<0.01		

**Table 5 jcm-10-01310-t005:** Pregnancy rate per patient in relation to baseline DFI. All couples received between 1–3 IUI treatments. The pregnancy rate was significantly higher for couples where the male DFI was ≤10 at baseline.

	N Patients	N Pregnancies	(%)	*p*-Value	Crude OR (95% CI)	Adjusted OR (95% CI)
**Pregnancy Rate**						
Total	146	46	31.5			
DFI ≤ 10 at baseline	55	25	45.5		0.36 (0.18–0.74)	0.30 (0.11–0.78)
DFI > 10 at baseline	91	21	23.1	<0.005		
Female age < 35	120	40	33.3			
DFI ≤ 10 at baseline	47	23	48.9		0.31 (0.14–0.70)	0.26 (0.08–0.78)
DFI > 10 at baseline	73	17	23.3	<0.005		
**Live Birth Rate**						
Total	142	29	20.4			
DFI ≤ 10 at baseline	53	18	34		0.27 (0.12–0.64)	0.14 (0.04–0.49)
DFI > 10 at baseline	89	11	12.4	<0.005		
Female age < 35	117	28	23.9			
DFI ≤ 10 at baseline	46	17	37		0.31 (0.13–0.75)	0.14 (0.04–0.51)
DFI > 10 at baseline	71	11	15.5	<0.01		

**Table 6 jcm-10-01310-t006:** DFI can in some males fluctuate. The males are divided into three groups. DFI ≤ 10: a minimum of two samples with a DFI ≤ 10, DFI > 10: a minimum of two samples with a high DFI > 10, or mixed: minimum one sample with low DFI ≤ 10 and 1 sample with high DFI > 10.

	N Patients	N Pregnancies	(%)	*p*-Value
**Pregnancy Rate**				
Total				
DFI ≤ 10	38	14	36.8	
DFI > 10	33	8	24.2	
mixed	55	18	32.7	0.51
**Live birth Rate**				
Total				
DFI ≤ 10	38	11	29	
DFI > 10	34	4	11.8	
mixed	54	15	27.8	0.15

## Data Availability

3rd Party Data. Restrictions apply to the availability of these data. Data was obtained from Aagaard/Bay Clinic and are available from the authors with the permission of Aagaard/Bay Clinic.

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
