# Peer review of "DNA Fragmentation in Human Spermatozoa and Pregnancy Rates after Intrauterine Insemination. Should the DFI Threshold Be Lowered?"

_jcm, 2021, doi:10.3390/jcm10061310_

Round 1

Reviewer 1 Report

General:

Authors evaluated relationship between initial DFI values of 146 patient couples undergoing IUI and pregnancy rate and live birth rate. They found if baseline pregnancy rate (PR)/cycle was 9.9% DFI was > 10 and 21.7% if baseline DFI was ≤ 10. The live birth rate (LBR)/cycle was 5% if baseline DFI was > 10 and 14.2% if baseline DFI was ≤ 10. PR/patient and LBR/patient were also significantly better if baseline DFI was ≤ 10.

They concluded that DFI > 10 could advice against timed coitus and non-stimulated IUI cycles and DFI measurement before treatment can be used to predict PR and LBR by IUI.

Major:

SCSA® is the pioneer method and most widely used technique for DFI measurement. Authors measured the DFI by an in-house flow cytometry (ref 5). They provided data concerning reagents, solutions and utensils in the Appendix A.

It is better to add some information concerning relationship between these two methods in M&M section.

Minor:

line 51   meta-analysis by Sugihara et al (2019).

          meta-analysis by Sugihara et al (2020).

(listed in Epub 2019 Oct 22)

Table 1

Basic characteristics for all couples and for couples with DFI information, and baseline semen evaluation should be listed on separate tables

Author Response

REVIEWER 1

General:

Authors evaluated relationship between initial DFI values of 146 patient couples undergoing IUI and pregnancy rate and live birth rate. They found if baseline pregnancy rate (PR)/cycle was 9.9% DFI was > 10 and 21.7% if baseline DFI was ≤ 10. The live birth rate (LBR)/cycle was 5% if baseline DFI was > 10 and 14.2% if baseline DFI was ≤ 10. PR/patient and LBR/patient were also significantly better if baseline DFI was ≤ 10.

They concluded that DFI > 10 could advice against timed coitus and non-stimulated IUI cycles and DFI measurement before treatment can be used to predict PR and LBR by IUI.

Major:

SCSA® is the pioneer method and most widely used technique for DFI measurement. Authors measured the DFI by an in-house flow cytometry (ref 5). They provided data concerning reagents, solutions and utensils in the Appendix A.

It is better to add some information concerning relationship between these two methods in M&M section.

 The information from Appendix 1 has been included in the M&M section alongside a description on the relationship between the two

Minor:

line 51   meta-analysis by Sugihara et al (2019).

          meta-analysis by Sugihara et al (2020).

(listed in Epub 2019 Oct 22)

This has been corrected.

Table 1

Basic characteristics for all couples and for couples with DFI information, and baseline semen evaluation should be listed on separate tables

This has been corrected so the two tables are separated.

Reviewer 2 Report

This is a cohort study where authors tried to evaluate the the value of DFI determined in a semen analysis collected before fertility treatment in the outcome of IUI. They concluded that analysis for DFI performed before treatment provides information about PR and LBR after IUI. There are several points to be revised.

Mild improvements in language and grammar are necessary.

The abstract section should have a more structured format, including the reporting of the methodology.

The rationale of the study should be given through more enriched reporting of the existing literature.

Was the protocol of the study published in any database? Please provide it.

Ι would suggest authors to provide with a sample size power calculation.

I would recommend authors to analyze how they dealt with selection biases; even this is a cohort study, allocation reporting of the patients has to be reported (e.g. 1 to 1 principle).

Ι also would suggest final results to be expressed by the final outcome (LBR) and authors to include an adverse effect as well, e.g. miscarriage, cancellation, OHSS. Thus, the final conclusion might be slightly different.

Author Response

REVIEWER 2

This is a cohort study where authors tried to evaluate the value of DFI determined in a semen analysis collected before fertility treatment in the outcome of IUI. They concluded that analysis for DFI performed before treatment provides information about PR and LBR after IUI. There are several points to be revised.

Mild improvements in language and grammar are necessary.

Linguistic improvements have been done.

The abstract section should have a more structured format, including the reporting of the methodology.

The abstract has been structured with more focus on methodology.

The rationale of the study should be given through more enriched reporting of the existing literature.

A section concerning the necessity of the study has been included in the introduction.

Was the protocol of the study published in any database? Please provide it.

The study is approved by the Regional Scientific Ethics Committee (S-20140212, Regionshuset, Damhaven 12, Vejle, Denmark). Informed consent forms have been collected for all participants before collection. All data has been collected in the RedCap database in order to meet requirements for data protection.

Ι would suggest authors to provide with a sample size power calculation.

A sample size calculation has been added.

I would recommend authors to analyze how they dealt with selection biases; even this is a cohort study, allocation reporting of the patients has to be reported (e.g. 1 to 1 principle).

We have carefully discussed potential selection bias. Although selection bias cannot be completely ruled out, but selection bias is less likely. IUI treatment in Denmark is free of charge for all citizens. The patients are referred to this clinic from their doctors and the patients did not go through any selection process. All patients (N=236) from the clinic were initially included in the study. However, we are aware of that 25 patients who did not initiate their fertility treatment was excluded. The exclusion is less likely to lead to selection bias because the fact that the reason of not initiating the treatment were most often due to the couple divorcing, a relocation or other events in their life e.g. new job, which results in a decision to postpone the fertility treatment. Of the 211 remaining couples, 65 had a missing baseline DFI. This is most often due to incidences in the laboratory, where it had not been possible to collect sample for the analysis. It is most likely that the missing is random. Therefore, we consider selection bias unlikely to be an issue.

The issue has been addressed under Strength and limitations.

Ι also would suggest final results to be expressed by the final outcome (LBR) and authors to include an adverse effect as well, e.g. miscarriage, cancellation, OHSS. Thus, the final conclusion might be slightly different.

LBR has been added in the discussion section. A description on miscarriage rate for LBR has been included. We do not have information about OHSS and cancelations.